# Cocrystals of Praziquantel with Phenolic Acids: Discovery, Characterization, and Evaluation

**DOI:** 10.3390/molecules27062022

**Published:** 2022-03-21

**Authors:** Shiying Yang, Qiwen Liu, Weiwen Ji, Qi An, Junke Song, Cheng Xing, Dezhi Yang, Li Zhang, Yang Lu, Guanhua Du

**Affiliations:** 1Beijing City Key Laboratory of Polymorphic Drugs, Center of Pharmaceutical Polymorphs, Institute of Materia Medica, Chinese Academy of Medical Sciences and Peking Union Medical College, Beijing 100050, China; ysy@imm.ac.cn (S.Y.); liuqw1995@163.com (Q.L.); jwwen0804@163.com (W.J.); a17861121095@163.com (Q.A.); xingc@imm.ac.cn (C.X.); 2Beijing City Key Laboratory of Drug Target and Screening Research, National Center for Pharmaceutical Screening, Institute of Materia Medica, Chinese Academy of Medical Sciences and Peking Union Medical College, Beijing 100050, China; smilejunke@imm.ac.cn (J.S.); dugh@imm.ac.cn (G.D.); 3Laboratory of Xinjiang Uygur Medical Research, Xinjiang Institute of Materia Medica, Urumqi 830004, China

**Keywords:** cocrystal, praziquantel, phenolic acid, solubility, evaluation

## Abstract

Solvent-assisted grinding (SAG) and solution slow evaporation (SSE) methods are generally used for the preparation of cocrystals. However, even by using the same solvent, active pharmaceutical ingredient (API), and cocrystal coformer (CCF), the cocrystals prepared using the two methods above are sometimes inconsistent. In the present study, in the cocrystal synthesis of praziquantel (PRA) with polyhydroxy phenolic acid, including protocatechuic acid (PA), gallic acid (GA), and ferulic acid (FA), five different cocrystals were prepared using SAG and SSE. Three of the cocrystals prepared using the SAG method have the structural characteristics of carboxylic acid dimer, and two cocrystals prepared using the SSE method formed cocrystal solvates with the structural characteristics of carboxylic acid monomer. For phenolic acids containing only one phenolic hydroxyl group (ferulic acid), when preparing cocrystals with PRA by using SAG and SSE, the same product was obtained. In addition, the weak molecular interactions that were observed in the cocrystal are explained at the molecular level by using theoretical calculation methods. Finally, the in vitro solubility of cocrystals without crystal solvents and in vivo bioavailability of PRA-FA were evaluated to further understand the influence on the physicochemical properties of API for the introduction of CCF.

## 1. Introduction

According to the World Health Organization, more than 250 million people worldwide are infected with schistosomiasis, and PRA is the most effective and widely used drug of choice for treating schistosomiasis [1,2,3]. However, PRA has the disadvantages of poor solubility and large dosage [4,5]. Because PRA has a very low water solubility (0.04 g/100 mL), a higher dose for clinical application could result in increased risks of adverse reactions and drug resistance, which may cause an enormous disaster for millions of people in the world if resistant parasites are produced and widely spread [6]. In addition, the larger size of praziquantel tablets is a challenge for children. Therefore, it is very important to improve the water solubility of praziquantel to reduce the dosage of praziquantel for better development of pediatric drug use and to reduce the probability of drug resistance and adverse reactions, as well as to protect global public health security [7].The solubility of praziquantel can be improved by preparing praziquantel solid dispersions, lipid nanoparticles or cyclodextrin inclusion compounds, but are limited by instability during preparation or storage. Considering the advantages of cocrystal technology in improving the physical and chemical properties of compounds, many researchers have carried out related studies on PRA [8,9,10,11]. At present, 27 cocrystals of PRA have been reported in the Cambridge Structural Database (CSD), and based on the analysis of their structural information, the cocrystal coformers (CCFs) of PRA were all carboxylic acid compounds or compounds with polyphenol hydroxyl groups. Therefore, in this study, natural phenolic acid compounds protocatechuic acid (PA), gallic acid (GA), and ferulic acid (FA), which have slightly better solubility, were selected as CCFs of PRA for cocrystal study to improve its solubility and bioavailability.

We previously reported the cocrystals of PRA with isoflavone compounds [12], and benzoic acid compounds [13] and their solubilities have been improved to varying degrees. In the present study, in the cocrystal synthesis of PRA with polyhydroxy phenolic acid, including PA, GA, and FA, five different cocrystals were prepared using SAG and SSE, and the preparation process of these cocrystals resulted in the following findings: (1) different cocrystals may be obtained with different degrees of solvent molecule involvement; and, (2) without the interaction sites to solvent molecules, the same cocrystal can usually be obtained using SAG and SSE. Various analytical techniques have been used for the solid-state characterization of these cocrystals. We also used theoretical calculation [14,15] to reveal the mechanism for this process. The solubilities of PRA in the cocrystals without crystal solvents were determined, and this parameter was significantly improved. Finally, we selected the cocrystal PRA-FA, which can be prepared using both methods to evaluate the biological activity in vivo. The results show that the exposure amount of PRA in vivo was obviously increased.

## 2. Results and Discussion

### 2.1. SXRD Analysis

Three of the cocrystals obtained single crystals, and their structural details were revealed by X-ray crystallography. These cocrystal structures have been deposited to CSD with deposition numbers of 2133511, 2133510, and 2133509. Both cocrystals PRA-PA-ACN and PRA-GA-ACN belonged to the monoclinic I 2/a space group, and, considering the structural similarity of CCF, the crystal cell parameters of the two cocrystals were also very similar. The asymmetric unit of PRA-PA-ACN and PRA-GA-ACN contains one PRA, one PA or GA, and one ACN molecule (left in Figure 1a,b). Cocrystal PRA-FA belonged to the monoclinic P 2_1_/n space group. The asymmetric unit of PRA-FA consisted of one PRA and one FA molecule (left in Figure 1c). Molecular packing maps in a unit cell are shown in the right part of Figure 1, where API is colored in green, CCF is colored in orange, and ACN is colored in magenta. The main crystallographic data of the three novel cocrystals are summarized in Table 1.

In these cocrystals, the two oxygen atoms of the carbonyl group in PRA and nitrogen atom in ACN acted as hydrogen bond acceptors, and the hydrogen atom of the carboxyl or phenolic hydroxyl group in the CCFs were hydrogen bond donors. The hydrogen bond formation mode and parameters are listed in Table 2. Figure 2 and Figure 3 show the hydrogen bond schemes and the corresponding contour map of the electron density difference of different interactions in PRA-PA-ACN, PRA-GA-ACN, and PRA-FA, respectively. As shown in Figure 3 and Figure 4, in the cocrystals PRA-PA-ACN and PRA-GA-ACN, phenolic acid formed hydrogen bonds with PRA and solvent in the form of monomer, and, in cocrystal PRA-FA, phenolic acid first formed dimer and then interacted with PRA by hydrogen bonding.

### 2.2. PXRD Analysis

The theoretical PXRD patterns calculated from SXRD data can be used as standard patterns to determine whether the prepared samples are cocrystals or not by comparison with experimental PXRD patterns. As shown in Figure 4, the experimental PXRD patterns matched well with the calculational ones, indicating that the obtained samples could be used in other characterization experiments.

The cocrystals PRA-PA and PRA-GA were obtained using the SAG method. As shown in Figure 5, compared with the PXRD patterns of API and CCFs, the PXRD patterns of the cocrystals showed significant differences in the number, intensity, and topological profile of the diffraction peaks, and these differences proved the formation of a new phase. In addition, the positions and peak intensities of the main intensity peaks of PRA-PA-ACN and PRA-GA-ACN in the PXRD patterns were relatively similar, while those of PRA-PA and PRA-GA in the PXRD patterns were relatively similar, but the former differ from the latter. This finding was obtained because the structures of CCF PA and GA are very similar and only differ in the 5-position hydroxyl group. Hence, the crystal stackings of PRA forming cocrystals with PA and GA are very similar, regardless of whether the formed cocrystals contain acetonitrile or not. Moreover, the crystal stacking patterns of PRA-PA and PRA-GA were also very similar, as evidenced by the characteristics of their endothermic peak values of DSC diagrams.

### 2.3. Thermal Analysis

The endothermic peak values in DSC curves (Figure 6) of the three cocrystals without crystal solvent were between that of API and CCF, and no obvious correlation was observed with the endothermic peak values of CCF. The two cocrystals’ acetonitrile solvates showed lower endothermic peaks, which belonged to crystal solvents. TG results show that approximately 7.97% and 7.83% of the weight loss occurred in cocrystal PRA-PA-ACN and PRA-GA-ACN, which were consistent with the theoretical weight loss (8.09% and 7.84%), confirming that approximately one acetonitrile molecule is present in the corresponding cocrystals. In addition, considering that the endothermic peaks of the two cocrystals were not observed after losing the solvents, the lattices of cocrystals should be destroyed following the loss of crystal acetonitrile and were transformed to amorphous because they failed to rearrange and pack immediately. We placed the samples of PRA-PA-ACN and PRA-GA-ACN in drying oven at 90 and 110 °C, and the products were named products A and B, respectively. The PXRD patterns of the samples showed characteristics of amorphous structure in Figure 5f,g, supporting the previous findings. Therefore, solvent molecules are important for maintaining the three-dimensional structure of cocrystal solvates, and the spatial arrangement of PRA-PA-ACN and PRA-GA-ACN differs from that of PRA-PA and PRA-GA.

### 2.4. IR Analysis

IR spectroscopy is a powerful tool for identifying cocrystal formation and distinguishing different cocrystal states, especially for cocrystals with specific structural fragments. In the present work, by using different preparation methods, different cocrystals were obtained, although the same solvent was used as the medium. The characteristic absorption peaks of the cocrystal of PRA with phenolic acids with and without crystal solvent were found by IR spectroscopy, in which, in the IR spectrum of acetonitrile solvates, the characteristic absorption peak is that from the C=O stretching vibration of conjugated carboxylic acid monomer at 1716 cm^−1^. In the IR spectrum of cocrystals without crystal solvent, the characteristic absorption peaks belonging to the C=O stretching vibration of conjugated carboxylic acid dimer were observed at 1667, 1686, and 1686 cm^−1^. In Figure 7, the green band represents the C=O stretching vibration peak of conjugated carboxylic acid monomer, and the orange band represents the C=O stretching vibration peak of conjugated carboxylic acid dimer.

### 2.5. Theoretical Computation

#### 2.5.1. Interaction Energy

The results of intermolecular interaction energy between Different Components in the Cocrystals are listed in Table 3. In cocrystal solvates PRA-PA-ACN and PRA-GA-ACN, the interaction energies of the carboxylic hydroxyl group of PA and GA (ring colored by cyan) with a carbonyl group on position 4 of PRA (ring B colored by green) were −16.39 and −16.05 kcal/mol, respectively, which do not differ much from each other; and the interaction energy between phenolic hydroxyl of PA and carbonyl group on position 4 of PRA was −13.95 kcal/mol, while the interaction energy between the phenolic hydroxyl of GA and carbonyl group on position 4 of PRA was −22.68 kcal/mol, which was much stronger. This phenomenon was mainly caused by the interaction simultaneously generated by two phenolic hydroxyl groups (on the position of 3 and 4) in GA. In addition, the interaction energy between PA and ACN (−4.57 kcal/mol) was lower than that between GA and ACN (−12.80 kcal/mol), supporting the fact that the desolvation temperature of PRA-PA-ACN (86.16 °C) was much lower than that of PRA-GA-ACN (104.88 °C).

In the PRA-FA, the interaction energy of FA dimer was−26.49 kcal/mol, which was larger than that of PRA-FA, PRA-PRA, that is, during the formation of PRA-FA, the connection between FA dimer was the most stable and easy to generate. For comparison, crystal information containing PA dimer and GA dimer was obtained from CSD (Identifier: BIJDON03 and AKISIY), and the interaction energies were calculated to be −21.99 and −35.22 kcal/mol, respectively. These two interactions were stronger than the other interactions, indicating that the formation of cocrystal with carboxylic dimer characteristic structure is dominant, which explains the formation of cocrystal PRA-PA, PRA-GA, and PRA-FA in the SAG process. In the solution system, a large number of solvent molecules are present, especially because of the presence of the interaction site (hydroxyl group on position 5 of PA and GA) with the solvent molecule. Therefore, the formation of PRA-PA-ACN and PRA-GA-ACN solvates became dominant. However, without the hydroxyl group on position 5 in FA, even in the solvent system, the cocrystal still formed with a characteristic dimer structure.

#### 2.5.2. Contour Maps of the Electron Density Difference

Figure 2 shows the contour maps of electron density difference of the two cocrystal acetonitrile solvates. Figure 3 shows the contour maps of the electron density difference of cocrystal PRA-FA. The solid red line represents the area where the electron density increased, and the dotted blue line represents the area where the electron density decreased. Based on the hydrogen bond interaction diagram formed by the hydroxyl and carbonyl groups (a, b, d and e), and hydroxyl and cyano groups (c and f) in Figure 2 and Figure 3, in the hydroxyl group as the hydrogen bond donor, the electron density around the hydroxyl hydrogen atom decreased sharply, especially toward the direction of the carbonyl oxygen atom. Moreover, the electron density of the hydroxyl oxygen atom toward the direction of the carbonyl group remarkably increased, whereas the electron density on both sides decreased slightly. In the carbonyl or cyanide group as the hydrogen bond acceptor, the electron density of the carbonyl oxygen or cyano nitrogen atom toward the direction of hydrogen atom decreased slightly, while the electron density on both sides increased slightly. The electron density between the hydroxyl hydrogen and carbonyl oxygen or cyano nitrogen atom significantly increased, indicating that the hydrogen bonds were formed.

#### 2.5.3. MEPS

The MEPS maps of API, CCFs, and solvent are shown as Figure 8. In PA, the global maxima value on MEPS near the hydroxyl group on position 4 was +63.00 kcal/mol, the local maxima value on MEPS near the hydroxyl group on position 7, which was smaller, was +52.25 kcal/mol, and the local maxima value near the hydroxyl group on position 5, which was the smallest, was +40.56 kcal/mol. The H bond interaction generated between the hydroxyl group on position 7 of PA and carbonyl group on position 7 of PRA (−40.60 kcal/mol) occurred pairwise by the interaction of the corresponding maxima and minima on the MEPS mentioned above. The H bond interaction generated between the hydroxyl group on position 7 of PA and carbonyl group on position 7 of PRA were formed similarly.

In GA, the global maxima value on MEPS near the hydroxyl group on position 4 was +63.06 kcal/mol, followed by the local maxima value on MEPS near the hydroxyl group on position 3 with a value of +53.79 kcal/mol. The local maxima value near the hydroxyl group on position 5, which was the smallest, was +42.22 kcal/mol. H bond interaction was generated between both of the hydroxyl groups on positions 3 and 4 of PA and carbonyl group on position 7 of PRA, in which the latter was stronger, thus supporting the results shown in the contour map of density difference (Figure 2). In this case, the increase of electron density between hydroxyl group on position 4 and carbonyl group on position 7 was remarkably higher than that between hydroxyl group on position 3 and carbonyl group on position 7. In addition, the local maxima value on MEPS near the hydroxyl group on position 7 was +52.52 kcal/mol, which was similar to PA, in which the H bond interaction generated between the hydroxyl groups on position 7 and carbonyl group on position 7 of PRA.

In the two cocrystal solvates mentioned above, the H bond occurred between ACN and PA or GA by the global maxima point (−39.26 kcal/mol) on MEPS near the cyano group interacted with the local maxima value near the hydroxyl group on position 5. No hydroxyl group was present on position 5 of FA. Hence, the solvate cannot be formed similar to PA and GA.

### 2.6. Solubility

Cocrystals without crystal solvents were selected to carry out solubility evaluation. Based on Figure 9, compared with PRA, the total amount of PRA dissolved from cocrystal PRA-FA was significantly improved in 420 min in the four dissolution systems. By using PRA as reference, the *f*_2_ values of the model independent similarity factor were calculated. The *f*_2_ values of cocrystal PRA-PA in four different systems were 43, 59, 43, and 33. The *f*_2_ values of cocrystal PRA-GA were 36, 47, 39, and 36; the *f*_2_ values of cocrystal PRA-FA were 44, 44, 27, and 44. Except for PRA-PA in the acetate buffer system (pH 4.5), the dissolution results of the three cocrystals in the four solvent systems differed remarkably from that in PRA, and the solubility of cocrystals was better.

### 2.7. In Vivo Evaluation

Considering that the solubility in vitro was greatly improved, we selected PRA-FA as the representative to evaluate the biological activity in vivo. The main pharmacokinetic parameters from plasma concentration versus time data of PRA and PRA-FA were calculated using DAS 3.0.7 software by a non-compartmental model; the results are listed in Table 4 and the plasma concentration–time curve of PRA and PRA-FA shows as Figure 10. The results showed that the pharmacokinetic process of PRA and PRA-FA were different in rats after intragastric administration in solid, and differences were observed among individuals. In comparison with PRA, cocrystal PRA-FA had higher in vivo exposure level (AUC(0–24 h)) and earlier peak time (Tmax), which was positively correlated with the in vitro solubility. However, the average residence time (MRT_(0–∞)_), maximum plasma concentration (Cmax), and half-life (t_1/2_) were basically consistent.

## 3. Materials and Methods

### 3.1. Materials and Chemicals

Raw PRA (purity: 98%) was obtained from Nanjing Pharmaceutical Factory Co., Ltd. (Nanjing, China). Phenolic acids, including protocatechuic acid (PA, purity: 99%), gallic acid (GA, purity: 99%), and ferulic acid (FA, purity: 99%), were obtained from Hubei Wande Chemical Co., Ltd. (Wuhan, China). All solvents used for crystallization were obtained from Sinopharm Chemical Reagent Beijing Co., Ltd. (Shanghai, China). The molecular structures of the compounds are shown in Figure 11.

### 3.2. Methods

#### 3.2.1. Preparation of Cocrystals

PRA-PA, PRA–GA, and PRA-FA: By using the SAG method, in a 1:1 molar ratio, 312.41 mg PRA and 154.12 mg PA, 170.12 mg GA, or 194.18 mg FA were weighed in a clean mortar and ground thoroughly with an approximate amount of acetonitrile (ACN) to obtain PRA-PA, PRA-GA, or a PRA-FA cocrystal powder sample. PRA-PA-ACN, PRA-GA-ACN, and PRA-FA: Approximately 200 mg PRA-PA, PRA-GA, or PRA-FA powder was weighed in a clean vial, added with acetonitrile until complete dissolution, and then filtered. By using the SSE method, block-shaped single crystals of PRA-PA-ACN, PRA-GA-ACN, and PRA-FA for SXRD were obtained.

#### 3.2.2. Single Crystal X-ray Diffraction (SXRD)

SXRD was performed using Rigaku MicroMax 002+ diffractometer (Rigaku Americas, the Woodlands, TX, USA) by using monochromatic CuKα radiation (λ = 1.54178) for supercritical XRD at 293 K. The experiment was measured under this condition, in which the collimating tube diameter was 0.30 mm, the CCD detector interval was 45 mm, the tube pressure was 45 kV, and the tube flow was 0.88 mA. CrystalClear software was used to restore and correct the collected single crystal data. Crystal structures were solved and refined through a direct method by using the Olex2 [16] and SHELXL [17,18] software. The least squares method was used to modify the structural parameters and distinguish the atomic types. The positions of all hydrogen atoms were obtained by geometric calculation and difference Fourier method.

#### 3.2.3. Powder X-ray Diffractometry (PXRD)

Powder X-ray diffraction data were collected using Rigaku D/Max-2550 powder X-ray diffractometer (Rigaku, Tokyo, Japan) with CuKα radiation of λ = 1.54178 Å at 40 kV and 150 mA. All samples, including API, CCFs, and new cocrystals, were analyzed between 2θ values of 3° and 40° with a step size of 0.02° and a scanning speed of 8°/min. The theoretical powder diagrams of the cocrystals were generated using the Mercury software [19,20] with starting and ending angles of 3–40°, step length of 0.02°, and half width of 0.15°.

#### 3.2.4. Differential Scanning Calorimetry (DSC) and Thermogravimetric Analysis (TGA)

DSC experiments were performed using a DSC instrument (Mettler Toledo, Greifensee, Switzerland) with a scanning rate of 10 °C min^−1^. The solid phases were weighed (3−5 mg) in aluminum pans with perforated lids. TGA experiments were performed using a TGA instrument (Mettler Toledo, Switzerland) at 30−500 °C at a scanning rate of 10 °C min^−1^. Solid samples (3−5 mg) were placed in aluminum oxide crucibles, and nitrogen was used as purge gas with a flow rate of 50 mL min^−1^. All DSC and TGA data were evaluated using STARe (Evaluation software 16.30) software.

#### 3.2.5. Fourier-Transform Infrared Spectroscopy (FT-IR)

The IR spectra of API, CCF, and cocrystals were obtained using a Spectrum 400 FTIR spectrometer (PerkinElmer, Waltham, United States of America), with the attenuated total reflectance method. Sixteen scans were performed, and the results were averaged for each spectrum. The scanning range of IR spectra was 4000–650 cm^−1^ at a resolution of 4.000 cm^−1^.

#### 3.2.6. Theoretical Computation

The intermolecular interaction energies of each H bond style of cocrystals were computed using the Gaussian 16 package [21] based on density functional theory. The B3LYP-D3/6-311G (d, p) level was employed for all the hydrogen atom geometry optimizations, while all the heavy atoms were observed at the original X-ray coordinates. The M06-2X/def2-TZVP level was used for single-energy calculations [22]. The Multiwfn 3.8 package [23] was used for contour maps of the electron density difference analysis and molecular electrostatic potential surface (MEPS) analysis.

#### 3.2.7. Powder Dissolution Experiments

To evaluate whether the cocrystals have solubility advantages over the API, we selected four solvent systems, including hydrochloric acid aqueous solution at pH 1.2, acetate buffer at pH 4.5, phosphate buffer at pH 6.8, and pure water at pH 7.0, to simulate different environments in the gastrointestinal tract [24,25]. The dissolution properties of API and the cocrystals were investigated by a basket method by using a RC12AD dissolution instrument (Tianda Tianfa Technology Co., Ltd., Tianjin, China). Accurately weighed samples (containing 30 mg PRA) were separately added to four different dissolution media (450 mL each). The experiment was measured at 37 °C and 100 rpm, and the samples were obtained and filtered at 0, 5, 15, 30, 45, 60, 90, 120, 180, 240, and 420 min. The contents of PRA were determined at 220 nm wavelength by high-performance liquid chromatography (HPLC, Agilent 1200, Santa Clara, CA, USA), and the dissolution contents were calculated using the external standard method. Mobile phase was a mixture of acetonitrile and water containing 0.2 vol % phosphoric acid aqueous solution at 50:50 vol %. The mobile phase was supplied at a constant flow rate of 1.0 mL/min. Chromatographic separation was performed in a Kromasil C18 column (250 mm × 4.6 mm, 5 μm) at 303 K. Precisely, 10 μL of the sample were injected.

#### 3.2.8. In Vivo Evaluation

To investigate whether the plasma concentration of cocrystal in rats increased compared with API, we performed a pharmacokinetics experiment of cocrystal PRA-FA in vivo [26,27]. SD rats were randomly divided into three groups with two rats in each group. The rats were fed under conventional feeding conditions, drank freely, fasted for 12 h, and were provided with PRA and cocrystal PRA-FA at 100 mg/kg by solid intragastric administration. At 15, 30, and 45 min and at 1, 1.5, 2, 3, 4, 6, 8, 12, and 24 h after administration, approximately 0.5 mL of blood from eye can thus be obtained to heparinize anticoagulant tubes, and centrifuged at 4000 rpm at 4 °C for 10 min, and the supernatant plasma was obtained for use. Approximately 100 µL of plasma were accurately absorbed and placed in a 1.5 mL centrifuge tube. Then, 10 µL of internal standard carbamazepine working solution (500 ng/mL) was added, followed by 1 mL of ethyl acetate (extraction solvent). The mixture was fully shaken for 3 min and centrifuged at 13,400 rpm for 10 min. Approximately 950 µL of upper solution were transferred into a centrifuge tube. The sample was dried with nitrogen at room temperature. A total of 100 µL of mobile phase (acetonitrile: water = 40:60, *v*/*v*) was added for redissolution, vortexed for 3 min, and centrifuged at 13,400 rpm for 10 min. Approximately 90 µL of the upper solution were placed in a lined tube for determination by a liquid chromatograph-mass spectrometer (LC/MS). Drug concentration was quantitatively analyzed based on the ratio of drug to internal standard peak area. The drug–hour curve was plotted with time as the abscissa and mean blood drug concentration as the ordinate.

## 4. Conclusions

Five cocrystals of PRA with phenolic acids, including protocatechuic acid, gallic acid, and ferulic acid, were studied in this paper. Three of these samples without crystal solvents were prepared using a solvent-assisted grinding approach, and the products were named PRA-PA, PRA-GA, and PRA-FA. These samples have the structural characteristics of carboxylic acid dimer fragments. Two of them with crystal solvents were prepared via SSE and named PRA-PA-ACN and PRA-GA-ACN, which have the structural characteristics of carboxylic acid monomer fragments. Various solid-state characterization techniques such as SXRD, PXRD, DSC, and IR were used to characterize these samples. The mechanism of cocrystal formation at the molecular level was discussed via theoretical calculation, including molecular interaction energy, electron density difference map, and MEPS. The effect of changes in molecular spatial arrangement caused by CCF intervention on physical properties was studied by selecting cocrystals without crystal solvents for in vitro and in vivo evaluation. The results show that the solubility of these cocrystals was significantly better than that of API. The Tmax and Cmax of PRA and cocrystal PRA-FA in vivo were basically the same, but the absorption degree of the latter was larger.

## Figures and Tables

**Figure 1 molecules-27-02022-f001:**
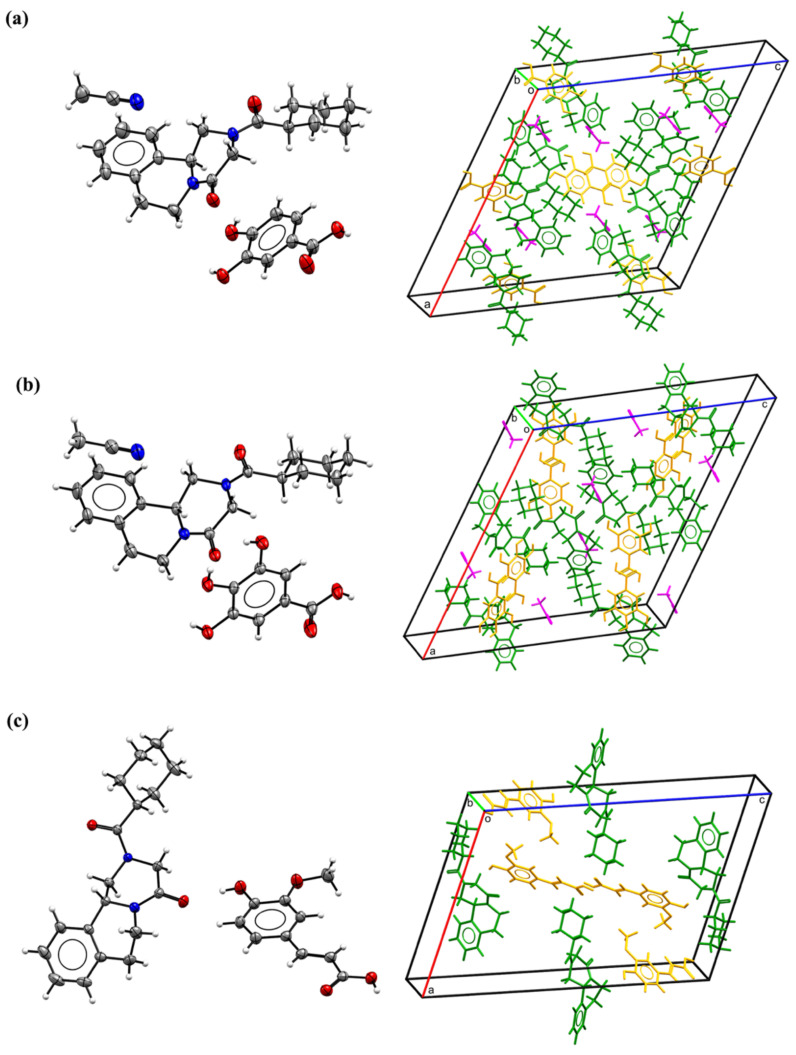
Thermal ellipsoid drawing (**left**) and molecular packing in a unit cell (**right**) of the cocrystals: (**a**) PRA-PA-ACN; (**b**) PRA-GA-ACN; (**c**) PRA-FA.

**Figure 2 molecules-27-02022-f002:**
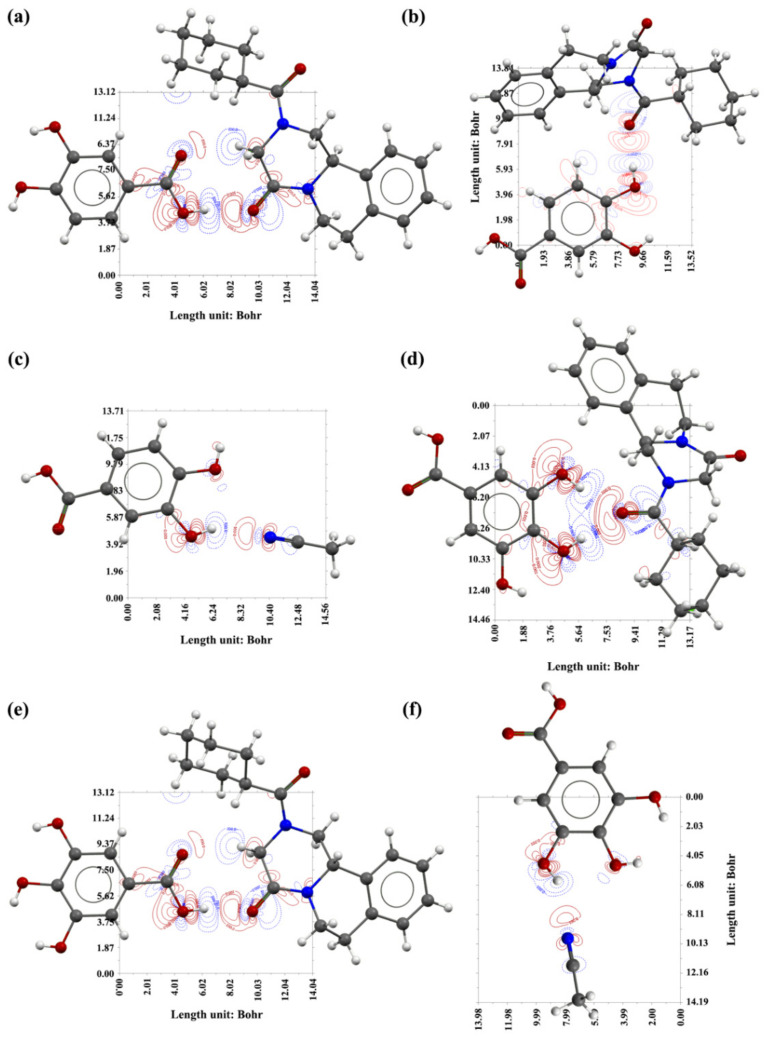
Hydrogen bond motifs and the corresponding contour map of the electron density difference of two cocrystal acetonitrile solvates. Hydrogen bond interaction: (**a**) PRA-PA-1; (**b**) PRA-PA-2; (**c**) PA-ACN; (**d**) PRA-GA-1; (**e**) PRA-GA-2; (**f**) GA-ACN.

**Figure 3 molecules-27-02022-f003:**
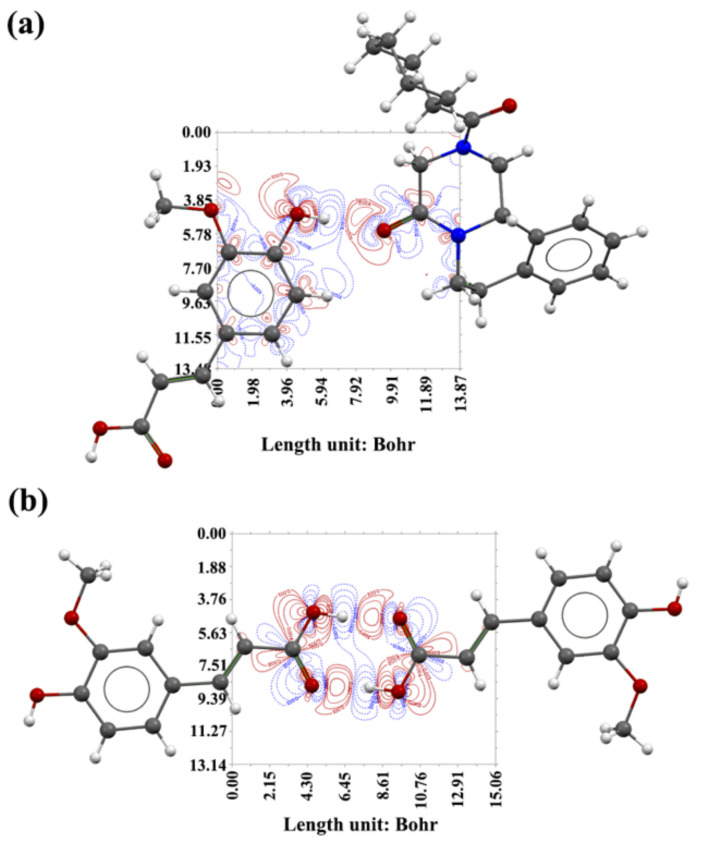
Hydrogen bond schemes and the corresponding contour map of the electron density difference of PRA-FA. Hydrogen bond interaction: (**a**) PRA-FA; (**b**) FA dimer.

**Figure 4 molecules-27-02022-f004:**
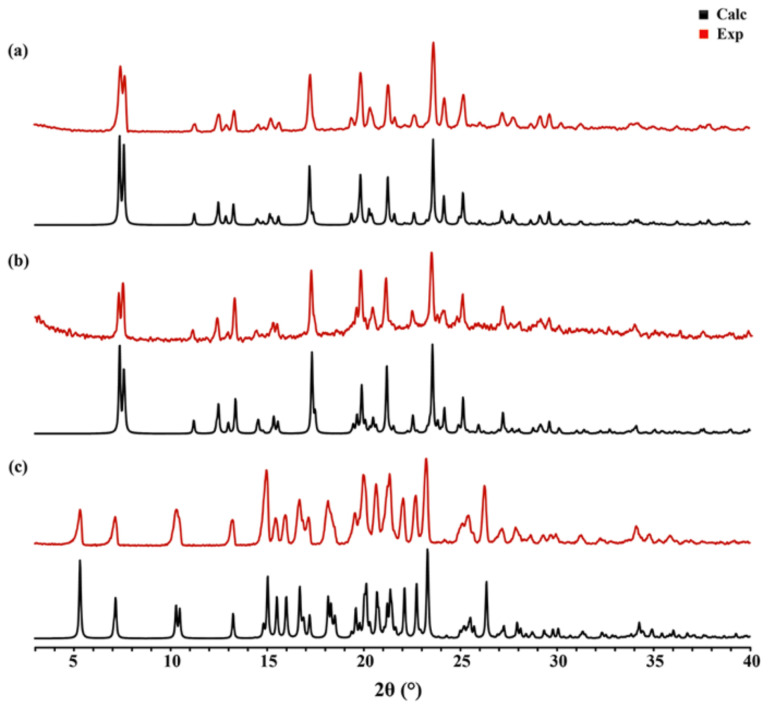
Calculational and experimental PXRD patterns of cocrystals: (**a**) PRA-PA-ACN; (**b**) PRA-GA-ACN; (**c**) PRA-FA.

**Figure 5 molecules-27-02022-f005:**
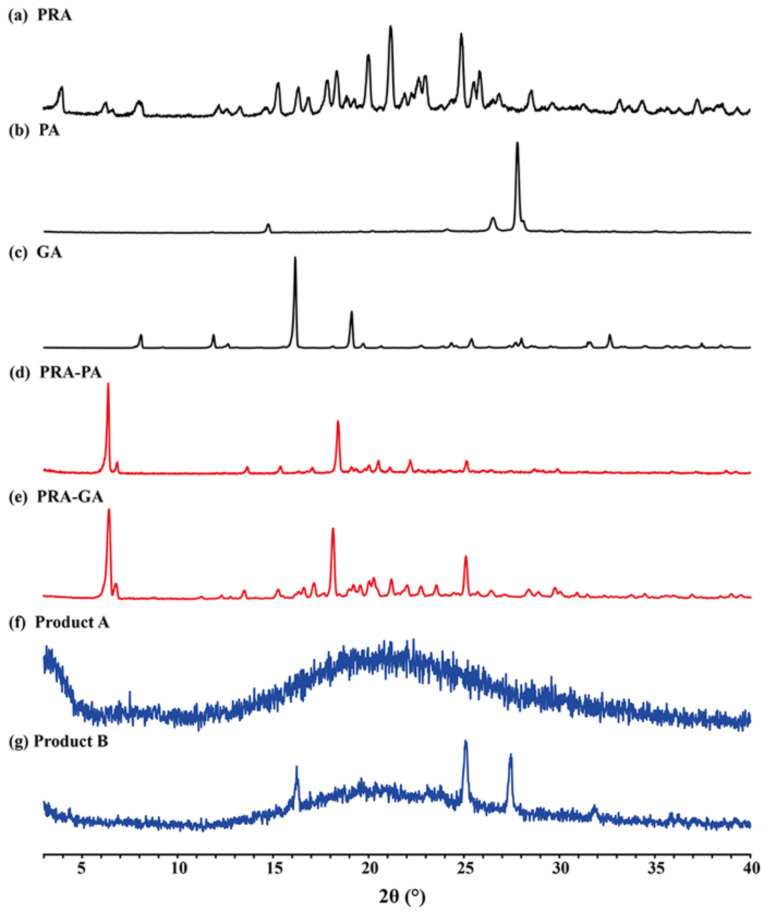
PXRD patterns of cocrystals PRA-PA and PRA-GA and the corresponding API and CCF (**a**–**e**) and thermal phase transformation products of PRA-PA-ACN (**f**) and PRA-GA-ACN (**g**).

**Figure 6 molecules-27-02022-f006:**
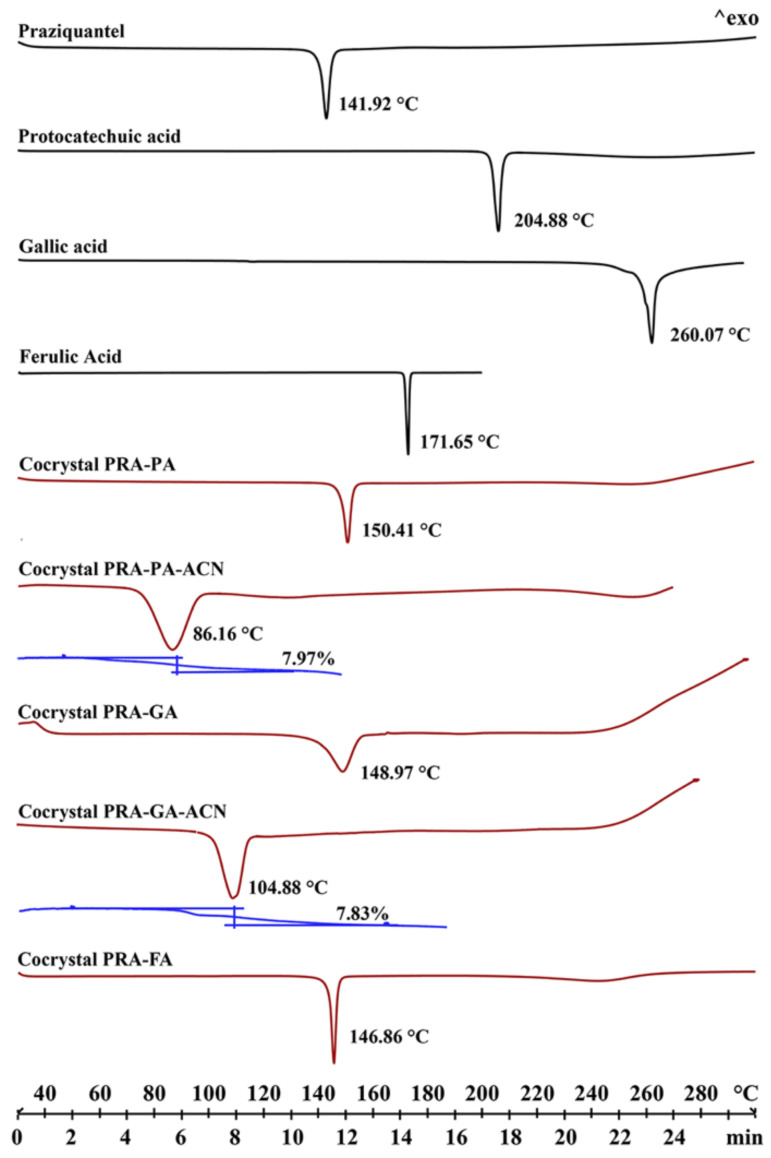
DSC profiles of the API, CCFs (in black), and the corresponding cocrystals (in red) and TG curves of solvates (in blue).

**Figure 7 molecules-27-02022-f007:**
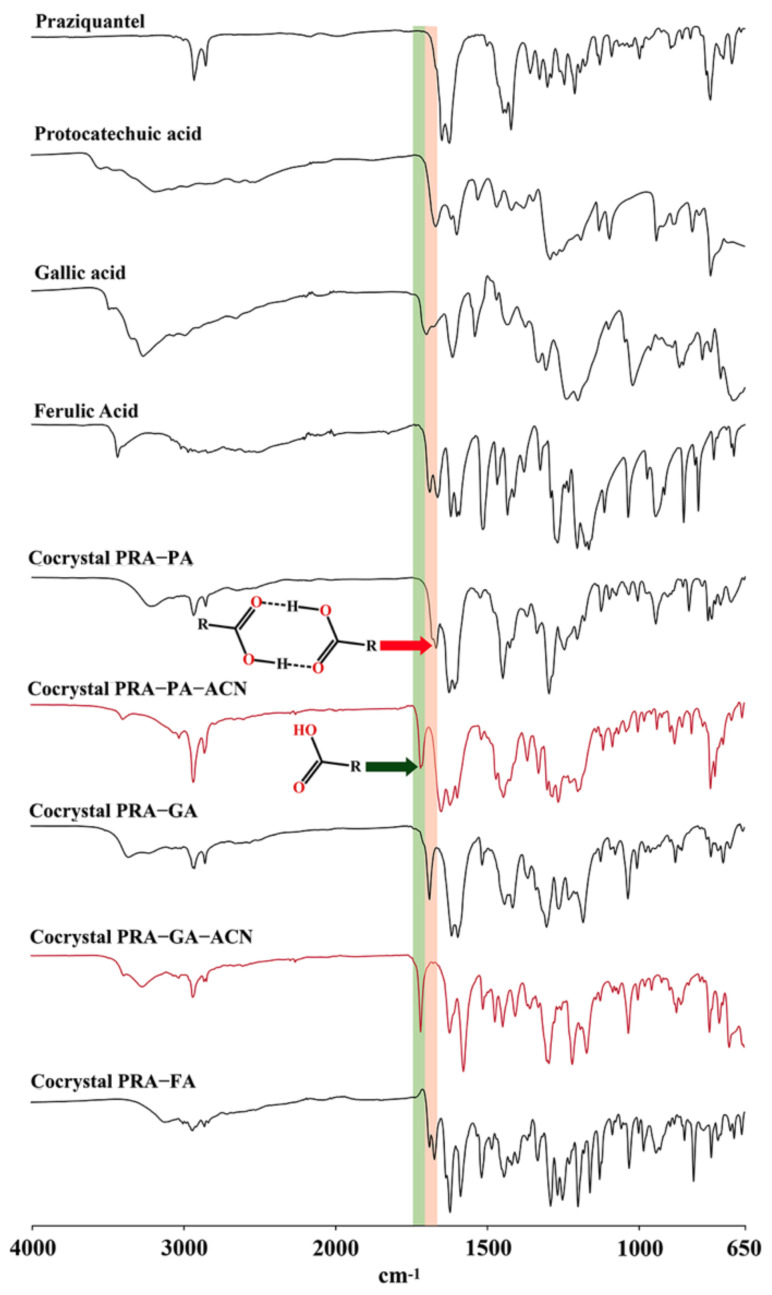
IR spectra of API, CCFs, the corresponding cocrystals (in black), and acetonitrile solvates (in red).

**Figure 8 molecules-27-02022-f008:**
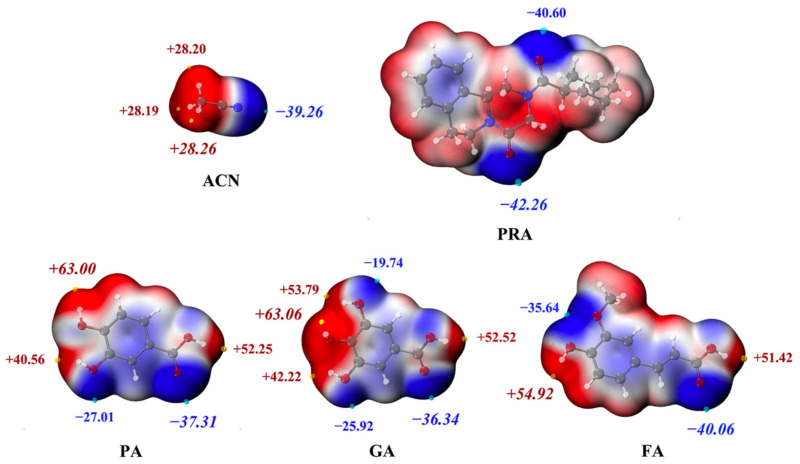
The MEPS maps of API, CCFs and solvent. Red represents the positive potential, and blue represents negative potential; the unit is kcal/mol.

**Figure 9 molecules-27-02022-f009:**
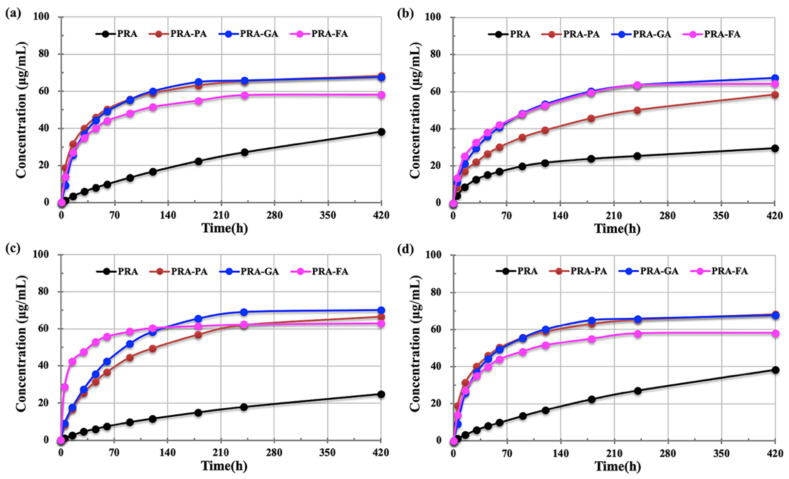
Solubility results of cocrystals of PRA-PA, PRA-GA and PRA-FA compared with PRA: (**a**) aqueous hydrochloric acid solution (pH = 1.2); (**b**) acetate buffer (pH = 4.5); (**c**) phosphate buffer (pH = 6.8); (**d**) water.

**Figure 10 molecules-27-02022-f010:**
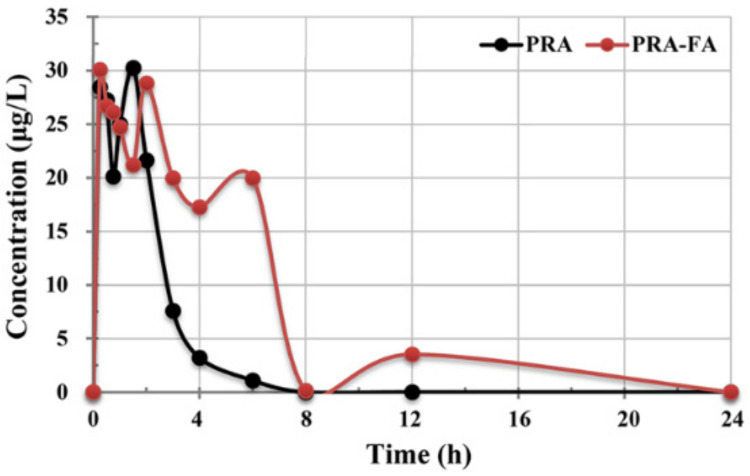
Plasma concentration–time curve of PRA and PRA-FA.

**Figure 11 molecules-27-02022-f011:**
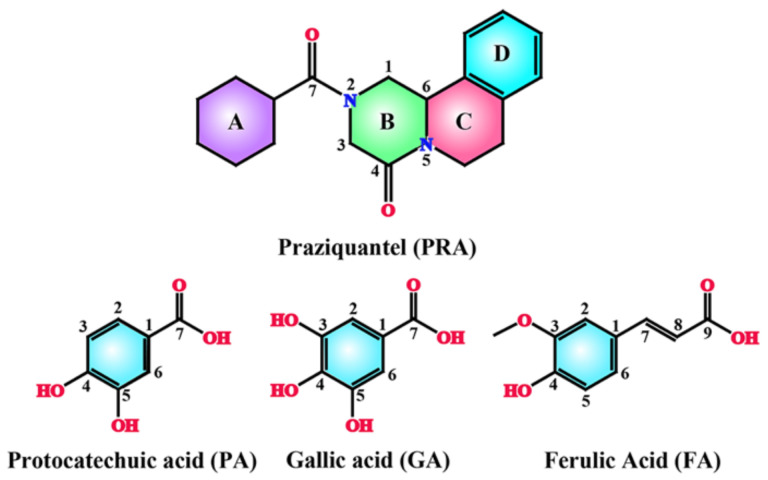
Molecular structures of API and CCFs. The rings with different colors are labeled to assist with the crystal structure description.

**Table 1 molecules-27-02022-t001:** Crystal cell parameters and structure refinement of the cocrystals.

	PRA-PA-ACN	PRA-GA-ACN	PRA-FA
Formula	C_28_H_33_N_3_O_6_	C_28_H_33_N_3_O_7_	C_29_H_34_N_2_O_6_
Crystal size (mm)	0.15 × 0.25 × 0.35	0.28 × 0.28 × 0.43	0.15 × 0.22 × 0.46
Description	block	block	block
Crystal system	monoclinic	monoclinic	monoclinic
Space group	I 2/a	I 2/a	P 2_1_/n
Unit cell parameters (Å, °)	a = 26.4615 (3)	a = 27.475 (1)	a = 17.7267 (19)
	b = 8.37690 (10)	b = 8.353 (1)	b = 5.8624 (7)
	c = 27.2338 (3)	c = 26.648 (1)	c = 26.209 (3)
	β = 118.541 (2)	β = 119.50 (1)	β = 108.058 (12)
Volume (Å^3^)	5303.17 (14)	5322.77 (7)	2589.5 (5)
Z	8	8	4
Density (g/cm^3^)	1.271	1.307	1.297
Theta range for data collection	3.695 < θ < 72.271	3.697 < θ < 72.311	3.412 < θ < 29.485
Independent reflections	5171	5191	5978
Reflections with I > 2σ(I)	4298	4990	3681
Completeness	99.8%	99.7%	99.2%
R (I > 2σI)	R = 0.051	R = 0.040	R = 0.055
	wR_2_ = 0.146	wR_2_ = 0.111	wR_2_ = 0.117
Goodness-of-fit on F^2^	1.072	1.075	1.018
Deposition Number	2133511	2133510	2133509

**Table 2 molecules-27-02022-t002:** Parameters of main hydrogen bonds for cocrystals.

Cocrystal	Interaction	D–H…A	D…A (Å)	<D–H…A (°)
PRA-PA-ACN	PRA-PA-1	^a^ O_1Y_–H_1Y_…O_1_	2.71	169.11
	PRA-PA-2	^b^ O_4Y_–H_4Y_…O_2_	2.62	174.69
	PA-ACN	^c^ O_3Y_–H_3Y_…N_1YJ_	2.98	151.24
PRA-GA-ACN	PRA-GA-1-1	^d^ O_4M_–H_4M_…O_2_	2.68	176.08
	PRA-GA-1-2	^e^ O_5M_–H_5M_…O_2_	2.68	174.82
	PRA-GA-2	^f^ O_2M_–H_2M_…O_1_	2.71	168.29
	GA-ACN	^g^ O_3M_–H_3M_…N_1YJ_	2.95	150.74
PRA-FA	PRA-FA	^h^ O_3_–H_3A_…O_1_	2.66	174.83
	FA dimer	^i^ O_6_–H_6_…O_5_	2.65	172.62

Symmetry Code: ^a^ −x + 1, −y + 1, −z + 2; ^b^ −x + 1, y−1/2, −z + 3/2; ^c^ x, y − 1, z; ^d^ −x + 1, y − 1/2, −z + 1/2; ^e^ −x + 1, y − 1/2, −z + 1/2; ^f^ −x + 3/2, −y + 1/2, −z + 1/2; ^g^ x, y − 1, z; ^h^ x + 1/2, −y + 5/2, z + 1/2; ^i^ −x, −y + 1, −z.

**Table 3 molecules-27-02022-t003:** Intermolecular interaction energy between different components in the cocrystals.

Interaction	Energy (kcal/mol)
PA-PRA-1	−16.39
PA-PRA-2	−13.95
PA-ACN	−4.57
PA-PA	−21.99
GA-PRA-1	−16.05
GA-PRA-2	−22.68
GA-ACN	-12.80
GA-GA	−35.22
PRA-FA	−11.13
FA-FA	−26.49
PRA-PRA	−7.71

**Table 4 molecules-27-02022-t004:** Parameters of main hydrogen bonds for cocrystals.

Parameter	PRA	PRA-FA
AUC_(0–24 h)_ (ng/mL·h)	73.67 ± 30.51	178.02 ± 190.75
MRT_(0–∞)_	3.02 ± 2.29	4.24 ± 1.99
T_max_ (h)	0.58 ± 0.38	3.00 ± 2.65
C_max_ (μg/L)	40.18 ± 14.66	38.93 ± 25.17
t_1/2_ (h)	2.03 ± 2.25	2.22 ± 1.55

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
