# Peer review of "Cocrystals of Praziquantel with Phenolic Acids: Discovery, Characterization, and Evaluation"

_molecules, 2022, doi:10.3390/molecules27062022_

Round 1

Reviewer 1 Report

This paper explores improvements in methods in chemical manipulation and preparation of praziquantel cocrystals with physicochemical properties of the API that will have better stability and consistency and bioavailability bordering on its mechanical properties. However, in the case of praziquantel, a formulation that will enhance the taste among oral formulation properties should be explored or discussed as part of this paper.

Consistency of PRA cocrystal is the main goal of this paper, but my personal interest in this paper would be to identify the relevance and utility of cocrystal research in improving the characteristics of praziquantel which though clearly indicated in the hypothesis and forms the basis for improving the physicochemical properties of PRA through co-crystallization and comparing the in vitro and in vivo properties of PRA, PRA-GA, PRA-FA may require more analysis and discussions.

The methods applied as indicated led to the determination and conclusions that the solubility of these cocrystals was significantly better than that of the current API but also forms an excellent basis for further work in this area of improving the pharmacokinetic properties of PRA.

To provide some more context to the paper the use of praziquantel may require some more details, and emphasis. PZQ is an important drug used in the treatment of schistosomiasis, one of the neglected tropical diseases and currently used in clinical and public health settings in many developing and lower-middle-income countries, and particularly as a public health tool for mass drug administration. Its safety and pharmacokinetic properties are well documented but like many other drugs improvements in its safety profile, especially the need to develop a well-tolerated pediatric formulation with appropriate safety characteristics for children as an important target for its use requiring an oral and liquid formulation of praziquantel have been proposed and advanced.

Questions on how do these changes impact the physicochemical structure and properties of praziquantel, that is what needed to change and how and improvements in consistency required merit this piece of research. This will in my estimation will provide more clarify and understanding of the hypothesis and what gains this research will elicit in the global health context?

Being significantly limited in my assessment and evaluation of this paper based on my depth of understanding and expertise in the physicochemical properties of praziquantel behind this research, the context of this piece of work becomes more significant to my assessment, and hope it holds for many others who would be reading this paper. The significance of this drug in addressing both global health needs and individual clinical care needs requires specific characteristics which should be clearly evaluated as an integral part of this paper.

The manuscript is well-written in clear and comprehensible language, the methods, analysis even within the limits of my personal constraints are plausible and make sense. Adequate referencing is made to support the methods, analysis, discussions and conclusions. I will recommend the paper for publication and rest assured that it will be well received and appreciated by the target audience.

Author Response

Point 1: This paper explores improvements in methods in chemical manipulation and preparation of praziquantel cocrystals with physicochemical properties of the API that will have better stability and consistency and bioavailability bordering on its mechanical properties. However, in the case of praziquantel, a formulation that will enhance the taste among oral formulation properties should be explored or discussed as part of this paper.

Consistency of PRA cocrystal is the main goal of this paper, but my personal interest in this paper would be to identify the relevance and utility of cocrystal research in improving the characteristics of praziquantel which though clearly indicated in the hypothesis and forms the basis for improving the physicochemical properties of PRA through co-crystallization and comparing the in vitro and in vivo properties of PRA, PRA-GA, PRA-FA may require more analysis and discussions.

 Response 1: First of all, thanks for your advices.The bitter taste and poor water solubility of praziquantel are the focus of its further development. In the screening process of praziquantel cocrystals, our research also selected CCFs based on this research idea. In the experiment, we selected sweeteners such as glycyrrhetinic acid and aspartame as CCF, but did not succeed. In this paper, natural phenolic acids with good water solubility and safety were selected, and the solubility improvement of PRA could be attributed to the introduction of CCF, which had good solubility. For molecular crystals, the increase of hydrophilic groups in the structure was conducive to the dissolution of cocrystals, such as the hydroxyl group and carboxyl group.

Point 2: The methods applied as indicated led to the determination and conclusions that the solubility of these cocrystals was significantly better than that of the current API but also forms an excellent basis for further work in this area of improving the pharmacokinetic properties of PRA.

To provide some more context to the paper the use of praziquantel may require some more details, and emphasis. PZQ is an important drug used in the treatment of schistosomiasis, one of the neglected tropical diseases and currently used in clinical and public health settings in many developing and lower-middle-income countries, and particularly as a public health tool for mass drug administration. Its safety and pharmacokinetic properties are well documented but like many other drugs improvements in its safety profile, especially the need to develop a well-tolerated pediatric formulation with appropriate safety characteristics for children as an important target for its use requiring an oral and liquid formulation of praziquantel have been proposed and advanced.

Questions on how do these changes impact the physicochemical structure and properties of praziquantel, that is what needed to change and how and improvements in consistency required merit this piece of research. This will in my estimation will provide more clarify and understanding of the hypothesis and what gains this research will elicit in the global health context?

Response 2: Thanks for your advices. According to your comments, the introduction has been modified to elaborate the purpose and significance of praziquantel cocrystal research.

Point 3: Being significantly limited in my assessment and evaluation of this paper based on my depth of understanding and expertise in the physicochemical properties of praziquantel behind this research, the context of this piece of work becomes more significant to my assessment, and hope it holds for many others who would be reading this paper. The significance of this drug in addressing both global health needs and individual clinical care needs requires specific characteristics which should be clearly evaluated as an integral part of this paper.

Response 3: Thanks for your advices. We have added some comments on the significance of this drug in addressing both global health needs and individual clinical care needs in the introduction section.

Reviewer 2 Report

This paper is described about the co-crystallization of PRA with phenolic acids such as gallic acid and ferulic acid, etc. Through various measurements such as X-ray structural analysis, the properties of the co-crystal are closely examined. I don't see any problem with these measurements, but it is not clear what is being calculated in the Theoretical Computation section, especially in the Interaction Energy section. This section should be deleted or the definition of Interaction energy should be clarified. Also, if the authors are simply estimating stabilization by hydrogen bonding, I think the -35.22 kcal/mol of GA-GA is a large estimate. Also, the "ring colored by cyan" and "ring B colored by green" lines 169-170, refers to the structural formula in line 277, and on first reading, it is not at all clear what it refers to.

I think the data in the In Vivo Evaluation section (2.7) has too much error. For example, the value of 178.02 ± 190.75 for the AUC of PRA-FA does not seem to be a very correct value because the error is much larger. However, I am not an expert in pharmacokinetics, so the opinions of other referees should be given priority in this section.

Author Response

Point 1: This paper is described about the co-crystallization of PRA with phenolic acids such as gallic acid and ferulic acid, etc. Through various measurements such as X-ray structural analysis, the properties of the co-crystal are closely examined. I don't see any problem with these measurements, but it is not clear what is being calculated in the Theoretical Computation section, especially in the Interaction Energy section. This section should be deleted or the definition of Interaction energy should be clarified. Also, if the authors are simply estimating stabilization by hydrogen bonding, I think the -35.22 kcal/mol of GA-GA is a large estimate. Also, the "ring colored by cyan" and "ring B colored by green" lines 169-170, refers to the structural formula in line 277, and on first reading, it is not at all clear what it refers to.

Response 1: First of all, thanks for your advices. Maybe we didn't make it clear, but the calculation of the intermolecular interaction energy is essential and indispensable. All data in Table 3, including -35.22 kcal/mol of GA-GA, were calculated based on density functional theory at the same level. The calculated results can qualitatively, not quantitatively compare the energy of each interaction in cocrystals, to gain more insight into the formation mechanism of cocrystals and to explain the reasons for the differences in the interaction pairs. The result of the interaction energy can explain why the desolvation temperature of PRA-PA-ACN (86.16 °C) was much lower than that of PRA-GA-ACN (104.88 °C) in the DSC pattern (figure 6), as well as explain the reason why PRA can form PRA-PA, PRA-GA, and PRA-FA instead of solvent compounds in SAG process. In addition, for a more visual presentation, the different ring are distinguished by different colors in Figure 11.

Thanks again for your comments. We have added descriptions in the revised manuscript to make the expression as clear as possible.

 Point 2: I think the data in the In Vivo Evaluation section (2.7) has too much error. For example, the value of 178.02 ± 190.75 for the AUC of PRA-FA does not seem to be a very correct value because the error is much larger. However, I am not an expert in pharmacokinetics, so the opinions of other referees should be given priority in this section.

Response 2: Thanks for your advices. We confirmed the values again and the final experimental results were indeed such results. In pharmacokinetic studies, such large deviations can occur because each rat has large individual differences, such as age, body weight, and metabolism of the drug, so it is normal for the values of the results to vary widely.
